# Intracranial Solitary Fibrous Tumour Management: A French Multicentre Retrospective Study

**DOI:** 10.3390/cancers15030704

**Published:** 2023-01-24

**Authors:** Marine Lottin, Alexandre Escande, Luc Bauchet, Marie Albert-Thananayagam, Maël Barthoulot, Matthieu Peyre, Mathieu Boone, Sonia Zouaoui, Jacques Guyotat, Guillaume Penchet, Johan Pallud, Henry Dufour, Evelyne Emery, Michel Lefranc, Sébastien Freppel, Houman Namaki, Edouard Gueye, Jean-Jacques Lemaire, Bertrand Muckensturm, Robin Srour, Stéphane Derrey, Apolline Monfilliette, Jean-Marc Constans, Claude-Alain Maurage, Bruno Chauffert, Nicolas Penel

**Affiliations:** 1Department of Oncology, Amiens University Hospital, 80054 Amiens, France; 2Department of Radiotherapy, Lille Oscar Lambret Centre, 59000 Lille, France; 3School of Medicine H. Warembourg, Lille University, 59000 Lille, France; 4Lab CRIStAL, Lille University, UMR 9189, 59655 Villeneuve d’Ascq, France; 5Department of Neurosurgery, Montpellier University Hospital, 34295 Montpellier, France; 6Institute of Functional Genomics, Montpellier University, 34000 Montpellier, France; 7Unit of French Brain Tumour DataBase, (Registre des Tumeurs de l’Hérault), Montpellier Cancer Insitute, 34000 Montpellier, France; 8Clinical Research Department, Oscar Lambret Centre, 59000 Lille, France; 9Department of Neurosurgery, Sorbonne University, La-Pitié-Salpêtrière University Hospital, 75651 Paris, France; 10Department of Neurosurgery, Lyon University Hospital, 69500 Bron, France; 11Department of Neurosurgery, Bordeaux University Hospital, 33076 Bordeaux, France; 12Department of Neurosurgery, Sainte-Anne University Hospital, 75014 Paris, France; 13Department of Neurosurgery, La Timone University Hospital, 13005 Marseille, France; 14Department of Neurosurgery, Caen University Hospital, 14033 Caen, France; 15Department of Neurosurgery, Amiens University Hospital, 80054 Amiens, France; 16Department of Neurosurgery, Saint-Pierre University Hospital, 97448 Saint Pierre, France; 17Department of Neurosurgery, Perpignan Hospital, 66046 Perpignan, France; 18Department of Neurosurgery, Limoges University Hospital, 87042 Limoges, France; 19Department of Neurosurgery, Clermont-Ferrand University Hospital, 63003 Clermont-Ferrand, France; 20Department of Neurosurgery, Orleans Hospital, 45067 Orleans, France; 21Department of Neurosurgery, Colmar Hospital, 68024 Colmar, France; 22Department of Neurosurgery, Rouen University Hospital, 76038 Rouen, France; 23Department of Medical Oncology, Lille University Hospital, 59000 Lille, France; 24Department of Radiology, Amiens University Hospital, 80054 Amiens, France; 25Department of Pathology, Lille University Hospital, 59000 Lille, France; 26Department of Oncology, Saint Quentin Hospital, 02321 Saint Quentin, France; 27Department of Medical oncology, Oscar Lambret Centre, 59000 Lille, France

**Keywords:** solitary fibrous tumour, hemangiopericytoma, intracranial, surgery, recurrence

## Abstract

**Simple Summary:**

Intracranial solitary fibrous tumours (iSFTs) are exceptional mesenchymal tumours with a high relapse rate. We aimed to analyse the clinical outcome at each stage of the disease. We carried out a multicentre retrospective study including 88 patients from 16 French centres. Gross tumour resection was found to be a factor for good prognosis and significantly reduced local recurrence without impacting overall survival. High-grade tumours were a factor for poorer PFS and LRFS. More than 40% of our patients experienced local recurrence and were mostly treated with surgery and radiotherapy. The first relapse is a turning point in iSFT evolution, with reduced recurrence latency over the course of the disease. The management of repeated recurrence and disseminated diseases is challenging; these situations should be treated, if feasible, with local techniques considering the poor efficiency of systemic treatments.

**Abstract:**

Background: Intracranial solitary fibrous tumour (iSFT) is an exceptional mesenchymal tumour with high recurrence rates. We aimed to analyse the clinical outcomes of newly diagnosed and recurrent iSFTs. Methods: We carried out a French retrospective multicentre (*n* = 16) study of histologically proven iSFT cases. Univariate and multivariate Cox models were used to estimate the prognosis value of the age, location, size, WHO grade, and surgical extent on overall survival (OS), progression-free survival (PFS), and local recurrence-free survival (LRFS). Results: Eighty-eight patients were included with a median age of 54.5 years. New iSFT cases were treated with gross tumour resection (GTR) (*n* = 75) or subtotal resection (STR) (*n* = 9) and postoperative radiotherapy (PORT) (*n* = 32, 57%). The median follow-up time was 7 years. The median OS, PFS, and LRFS were 13 years, 7 years, and 7 years, respectively. Forty-two patients experienced recurrence. Extracranial metastasis occurred in 16 patients. Median OS and PFS after the first recurrence were 6 years and 15.4 months, respectively. A higher histological grade was a prognosis factor for PFS (*p* = 0.04) and LRFS (*p* = 0.03). GTR influenced LRFS (*p* = 0.03). Conclusion: GTR provided benefits as a first treatment for iSFTs. However, approximately 40% of patients experienced relapse, which remains a challenging state.

## 1. Introduction

Intracranial solitary fibrous tumours (iSFTs) and hemangiopericytoma are rare primary tumours of the central nervous system (CNS) [1]. These mesenchymal tumours are considered sarcomas and represent less than 1% of primary CNS tumours and 2–5% of all meningeal tumours [1,2]. Their incidence in France is 0.061 per 100,000 inhabitants per year and 0.041 per 100,000 inhabitants worldwide [3]. ISFTs emerge from pericytes, a mural component of vessels that enables the function of the blood–brain barrier and the support of intracranial immunity [1,2]. Until recently, SFTs were distinguished from hemangiopericytoma (HPC) because of their difference in terms of aggressiveness. Even if SFTs were mostly considered benign tumours, some have a malignant evolution [1]. Conversely, HPC evolution may mimic benign tumours [1]. Thus, based on those characteristics and a common biological anomaly, i.e., the fusion of the NAB2-STAT6 gene arising from the chromosome 12q13 inversion, the 2016 WHO classification merged SFTs and HPC into the single entity “SFT/HPC” [4,5]. To align with the soft tissue nomenclature, this entity was renamed “SFT” in 2021.

Recurrences of iSFT are frequent, iterative, and more commonly located at the initial tumour site [6,7]. Because of their low occurrence, evidence for the general management of iSFTs still relies on few prospective data and retrospective studies [1,6,8,9]. Relapses are even less described, leading to heterogeneous practice between and within countries. Those relapses might also be metastatic with poor outcomes [10,11]. The aim of iSFT management must be to reduce the percentage of local and distant recurrences while considering practicable therapy [1,12]. Surgery is the universal mainstay of SFT treatment. A complete surgery or “gross tumour resection” (GTR) enables a significant gain in terms of survival and local control [1,2]. In addition, postoperative radiotherapy (PORT) provides good local control, especially after a “subtotal tumour resection” (STR) and high-grade cases. However, PORT’s impact on the overall survival (OS) of iSFTs is still discussed, leading to various approaches to management at international and national levels [1,10,13,14]. Given the difficulty of performing successive local interventions without considerable side effects, systemic treatments might be employed to treat local recurrences. Systemic treatments are mostly used in the metastatic stage, although the impact of survival remains inconclusive [8,11]. Thus, the treatment of local and distant relapses remains a serious source of concern, lacking sufficient guidelines.

Currently, very few studies have examined the survival outcome of iSFTs according to their treatment following diagnosis and relapse [9,15,16]. Firstly, we aimed to analyse iSFT outcomes at each stage of the disease in a large, multicentre, retrospective French cohort. Secondly, we aimed to analyse the associated clinicopathological factors to provide a better understanding and enhance future therapeutic decisions.

## 2. Materials and Methods

### 2.1. Case Selection

We collected the data of cases from the “French brain tumour database” (FBTDB) in collaboration with the French neuropathology network (RENOCLIP-LOC). The database combines information of patients with a confirmed pathological diagnosis of intracranial SFT, HPC, and anaplastic HPC in France from 2006 to 2015. Clinical and pathological records of the patients were retrospectively reviewed to extract the relevant clinical factors at 16 French centres (Amiens, Bordeaux, Caen, Clermont-Ferrand, Colmar/Strasbourg, Ile de la Réunion, Lille, Limoges, Lyon, Marseille, Montpellier, Orléans, Paris-La-pitié-salpêtrière, Paris-Sainte-Anne, Perpignan, and Rouen) from January 2006 to December 2015. The exclusion criteria were disease occurrence outside the inclusion period or at an extracranial level, a lack of histological proof of SFT, age <18 years at diagnosis, and a written refusal of consent to the study. Information on age, sex, clinical presentation of the disease, time to diagnosis, tumour location, tumour size, extent of resection, preoperative biopsy, preoperative embolization, pathology including immunohistochemistry, adjuvant radiotherapy, tumour recurrence or metastasis occurrence, and patient survival was collected.

Intracranial hypertension was defined by the association of headaches, nausea, and vomiting, and was distinguished from exclusive headaches in the clinical presentation of the disease. Tumour size corresponded to the maximal diameter measured by the radiologist on the preoperative radiologic sequences. Time to diagnosis was estimated as the period between the outbreak of evocative neurological symptoms and the date of the established histology. The extent of resection was classified as GTR or STR according to the postoperative MRI results. As meningiomas, GTR and STR are defined as Simpson grades I or II removal and grades III or IV removal, respectively [1,12]. Preoperative intervention through embolization guided with arteriography was conducted at the surgeon’s discretion. Tumour local recurrence was defined as the reappearance of the tumour within the cranial cavity or an increase in the size of the residual tumour according to RECIST1.1 criteria. Metastasis was defined as an extracranial appearance of SFT.

### 2.2. Pathology

All specimens of SFTs were confirmed by a pathologist with expertise in this disease. Specimens were graded according to the 2021 CNS WHO classification into grades 1, 2, or 3 [5]. According to these criteria, grade 1 is characterized by a highly collagenous, relatively low-cellularity spindle-cell lesion, previously diagnosed as SFT. Grade 2 corresponds to a more cellular, less collagenous tumour with plump cells, staghorn vasculature, and mitosis <5 per 10 high-power field (HPF), previously diagnosed as HPC. Grade 3 corresponds to ≥5 mitoses/10 HPF and/or the presence of necrosis, and it was previously named “anaplastic HPC”.

### 2.3. Statistical Analyses

Patient characteristics were described using the mean (standard deviation (SD)) or median (range) as appropriate for continuous variables and frequency (percentage) for categorical variables. Missing data were reported.

The median follow-up time was estimated by using the reverse Kaplan–Meier method, considering survival status at the end of the study as an event and death as a censored event. Survival analyses were performed using the Kaplan–Meier method. The survival rate, median, and confidence interval at 95% (95% confidence interval (95% CI)) were reported. Follow-up times were defined as the time from the date of the initial pathological diagnosis of SFT to the date of (1) death for OS; (2) recurrence or death if no recurrence occurred for progression-free survival (PFS); (3) local recurrence or death if no local recurrence occurred for local recurrence-free survival (LRFS). In cases of recurrence, follow-up times were defined as the time from the date of first recurrence of SFT to the date of (4) death for the second OS (OS2) and (5) the second recurrence or death if no second recurrence occurred for the second PFS (PFS2). Patients who remained alive or were lost-to-follow-up were censored.

The hazard ratio (HR) for recurrence and/or death (PFS) or for death (OS) associated with patients’ characteristics were estimated using Cox models—first, in the univariate analysis and then in the multivariate analysis adjusted for possible confounders. The initial multivariate model included all covariates with a *p*-value < 0.20 in univariate analyses except for adjuvant radiotherapy, which only concerned the subgroup of patients with the indication according to the literature [14]. Correlations between variables were searched. The backward selection gave the final multivariate model including only covariates with *p*-value < 0.05. Sensitivity multivariate analyses were performed with PORT as a covariate. All the point estimates were reported with their 95% CI. All tests were two-sided, and the threshold for statistical significance was set to *p* < 0.05.

The software used for the analyses was Stata version 17.0 (StataCorp. 2021. Stata Statistical Software: Release 17. College Station, TX: StataCorp LLC).

### 2.4. Ethical Approval

The study complied with the “reference methodology” MR004 adopted by the French Data Protection Authority (CNIL), and we checked that patients did not object to the use of their clinical data for the research purpose. Ethical local approval was obtained on 25 May 2022. The number of the ethical approval is N° 2022-013. All data were anonymized.

## 3. Results

### 3.1. Patient Characteristics

A query of the database revealed 179 patients with a primary iSFT from January 2006 to December 2015 in the 16 participating French centres. Ninety-one cases were excluded because they were diagnosed before 2006 or after 2015 (*n* = 37), had extracranial localization (*n* = 19), were missing proof of the disease (*n* = 28), had a revised histology by pathological review (*n* = 6), or were patients aged <18 years old (*n* = 1). Finally, we included 88 patients in the study (Figure 1).

There was a female predominance (*n* = 50, 57%), and the median age at diagnosis was 54.5 years (range: 19–88). The SFTs were mainly local (*n* = 84, 95%) and supratentorial (*n* = 71, 81%) at diagnosis. Tumours often had a large maximal diameter on brain MRI (≥5 cm for 32 (36%) patients-Table 1). All the clinical characteristics of the patients were similar regardless of tumour lateralization and are summarized in Table 1. 

### 3.2. Clinical and Pathological Presentation

The symptoms leading to diagnosis were related to the tumour location and consisted mainly of headaches (*n* = 41, 47%) and cognitive disorders (*n* = 37, 42%) (see Appendix A).

Seventy-seven tumours (70%) were graded according to the 2021 CNS WHO classification. There were 1 (1%), 25 (28%), and 51 (58%) grade one, two, and three iSFT cases, respectively. The 34 samples (39%) that were tested for the nuclear expression of STAT6 using immunohistochemistry were all positive (Table 1). The characteristics of the population with STAT6 status results were similar to those with an unknown status.

### 3.3. Initial Treatment

Eighty-seven (99%) patients underwent surgical resection and one (1%) had a biopsy without an additional resection. GTR and STR were achieved in 75 (85%) and 9 (10%) patients, respectively. The surgical excision (GTR or STR) status of the three other patients was unknown (Table 1). Only eight patients (9%) received preoperative embolization. After initial surgery, 32 patients (57%) received adjuvant radiotherapy. One of the 32 patients received exclusive radiotherapy after a single biopsy without resection. Systemic treatments were not employed at the initial stage (Table 1).

### 3.4. Survival Analyses

The median follow-up time was 7 years (range 0–16 years). Twenty-four patients had died by the retrospective analysis time, seven (29%) of whom had metastatic disease. The median OS time was 13 years (95% CI: 10–nonreached (NR)), with 1-, 5-, and 10-year OS rates of 93% (95% CI: 85–97), 85% (95% CI: 74–91), and 64% (95% CI: 48–76), respectively. The median PFS was 7 years (95% CI: 6–8 years) after the date of diagnosis, with 1-, 5-, and 10-year PFS rates of 87% (95% CI: 78–93), 66% (95% CI: 54–76), and 19% (95% CI: 9–32), respectively. The median LRFS was 7 years (95% CI: 6–8 years) after the date of diagnosis, with 1-, 5-, and 10-year LRFS rates of 87% (95% CI: 78–93), 68% (95% CI: 56–77), and 23% (95% CI: 11–36), respectively (Figure 2).

The median OS2 was 6 years (95% CI 4–NR), with 1-, 3-, and 5-year OS2 of 87% (95% CI 72–94), 75% (95% CI 57–86), and 53% (95% CI 31–71), respectively. The median PFS2 was 15.4 months (95% CI 9.9–32.4), with 1-, 3-, and 5-year PFS2 rates of 61% (95% CI 44–74), 33% (95% CI 18–49), and 10% (95% CI 2–26), respectively (Figure 2D,E).

### 3.5. Recurrences and Treatment at Recurrence

During the follow-up, 42 patients (48%) experienced at least one progression or recurrence and 16 (18%) patients had a distant recurrence. Among the 42 relapses, 35 (83%) were located at the initial site, 5 (12%) were distant, and 2 (5%) were simultaneously local and distant. The morphological and treatment characteristics of the first local recurrence are summarized in Table 2. Among the 37 patients presenting a local relapse, 17 (46%) received surgical treatment, of whom 10 (59%) underwent PORT. Exclusive radiotherapy was performed in 14 patients (38%) and systemic treatment in 2 (5%) (Table 2).

A second recurrence occurred in 25 patients, of whom 16 were local (64%) and treatment corresponded to surgery for 6 (38%) patients, radiotherapy for 6 (38%), and systemic treatment for 3 (19%) patients. The relapses were repetitive, and we observed fifth (*n* = 3), sixth (*n* = 2), and seventh (*n* = 1) recurrences in our series, which were all metastatic. Their treatment consisted of either surgery, radiotherapy, or systemic treatments. To our knowledge, no other solid or haematological cancer occurred during the time of surveillance.

### 3.6. Extracranial Metastasis

Sixteen patients (18%) developed extracranial metastases. The most frequent sites of extracranial metastases were the liver (five patients, 31%), bones (nine patients, 56%), and the lungs (six patients, 38%). Interestingly, only eight (50%) patients received a systemic treatment at this disseminated stage. The treatment consisted of chemotherapy (doxorubicin *n* = 2; temozolomide *n* = 2; doxorubicin + ifosfamide *n* = 1) in five (62.5%) patients and antiangiogenic targeted therapy in two patients (25%) (bevacizumab *n* = 1, pazopanib *n* = 1). One patient (12.5%) was treated with a combination of chemotherapy and vascular endothelial growth factor (VEGF) inhibitor (temozolomide and bevacizumab). A complete response was observed in five patients who underwent surgery or radiotherapy. The best response in the systemic group was a stable disease (see Appendix A).

### 3.7. Prognosis Factors

We observed that age was associated with poorer OS (HR = 1.04 (95% CI: 1.01–1.08); *p* = 0.01), PFS (HR = 1.03 (95% CI: 1.01–1.05); *p* = 0.008), and LRFS (HR = 1.04 (95% CI: 1.01–1.06); *p* = 0.002) (Table 3). Tumour topography (midline and left lateralization) was associated with worse OS in the multivariate analyses with HR = 2.69 (95% CI: 1.07–6.81), *p* = 0.03 for left tumours and HR = 8.29 (95% CI: 1.42-48.50), *p* = 0.03 for midline tumours, as compared with right-sided tumours (Table 3, Figure 3A). A higher histological grade was significantly associated with lower PFS and LRFS (HR = 2.14 (95% CI: 1.03–4.78; *p* = 0.04) and HR = 2.36 (95% CI: 1.08–5.16; *p* = 0.03), respectively) (Table 3).

We observed that treatment influenced the outcome. We observed a significantly lower LRFS in the STR group in the multivariate analyses with HR = 3.00 (95% CI: 1.09–8.24; *p* = 0.03) (Table 3, Figure 3D). However, performing STR was associated with significantly lower OS only in the univariate analysis with HR = 3.20 (95% CI: 1.04–9.83; *p* = 0.04) and tended to be associated with lower PFS (HR = 2.71 (95% CI: 0.99–7.41; *p* = 0.052)). Preoperative embolization was associated with a lower risk of PFS and LRFS in the univariate analysis, but this association was not significant in the multivariate analysis (HR = 0.08 (95% CI: 0.001–0.55; *p* = 0.08) and HR = 0.09 (95% CI: 0.001–0.60; *p* = 0.09), respectively). Adjuvant radiotherapy was significantly associated with greater PFS and LRFS in the univariate analysis (HR = 0.50 (95% CI: 0.26–0.97; *p* = 0.04) and HR = 0.56 [95% CI: 0.29–1.10]; *p* = 0.10)), respectively, but not with OS (HR = 0.88 (95% CI: 0.34–2.29); *p* = 0.80). Sensitivity multivariate analyses were performed with PORT as a covariate. We observed a significantly lower PFS with HR = 0.47 (95% CI: 0.24–0.91; *p* = 0.03), but the association was no more significant regarding LRFS.

## 4. Discussion

Intracranial SFTs are very rare mesenchymal tumours with a high risk of recurrence [9,15,17]. To the best of our knowledge, this was the largest clinical retrospective cohort describing data at the stage of diagnosis and recurrence in Europe and the USA. There are no established guidelines for the management of iSFTs, a hard-to-treat malignancy, especially at recurrence. Current evidence came from retrospective data and prospective studies with a limited number of cases [18]. We conducted a long-term study permitting the report of 42 patients in relapse and their characterization, which is usually difficult given the rarity of this entity [2,6,19]. Local treatments are often employed early and systemic treatment seems to present a restrained efficiency.

As reported by several authors, we observed that GTR provided a strong benefit in terms of tumour control as a first treatment for iSFTs [6,10,12,13,20]. Kim et al. reported a significant decrease in the 5-year recurrence risk in GTR cases compared to STR (20.8% vs. 72.7%; *p* = 0.006) cases [2]. However, the results of GTR on OS remain controversial. Consistent with our observation, Soyuer et al. found no association between survival and resection extent [10]. However, Rutkowski et al. observed an increased survival with GTR (13 vs. 9.7 years, *p* < 0.05), independent of the realisation of PORT [9].

Several studies have analysed the impact of PORT on iSFT recurrence and survival with conflicting results [7,14,21,22,23]. In our cohort, PORT seemed to be efficient only in terms of local control and PFS in patients who presented with an aggressive tumour or underwent an incomplete resection as, discussed in Stessin et al. Other authors, such as Coombs et al., even reported improved OS [7,12,14,21]. However, Xiao et al. found no association between PORT and survival or PFS irrespective of the quality of resection [24]. The low percentage of STR cases included in our cohort and missing data induced a lack of power that could explain our observations. Thus, our cohort could not engage the role of adjuvant radiotherapy in these situations, but the results remained promising. Comparison between different radiotherapy techniques and regimens or between stereotactic radiosurgery (SRS) and fractionated radiotherapy was not addressed in our series due to the small number of patients undergoing adjuvant SRS and missing data. Moreover, due to the retrospective nature of our cohort based on surgical specimens, no patient treated with exclusive SRS was included in the study. However, like meningiomas, this treatment strategy should be further explored.

Based on the rationale of the meningioma treatment, a preoperative embolization is sometimes conducted to perform a GTR and limit morbidity in these heavily vascularized tumours [19,25]. To date, its impact on clinical outcome has not been demonstrated in iSFTs. In our series, this technique tended to reduce recurrence. The nonsignificant association in the multivariate models might have been due to the lack of power because of the small number of patients.

Concerning survival predictive factors, age at diagnosis is heavily discussed [10,12,26]. However, as Wang et al., we found that a 1-year increase in age predicted a poor outcome [15]. We observe that the CNS WHO grade III was associated with poorer PFS and LRFS as Macagno et al. had described [4,20].

We reported an association between survival and tumour location. The results concerning midline locations should be interpreted with caution, given the very small number of patients (*n* = 4) with this type of location. To our knowledge, this information was not described in other studies of SFTs. Even if extra- and intra-axial tumours are not similar, some authors have described a poor prognosis in glioblastoma with central and left temporal localizations [27]. The impact of left lateralization on OS in our series might be explained by the greater preoperative risks due to the presence of language centres, frequently located on the left [12,13]. However, we did not observe a negative impact for tumours located at intrasinusal or cerebellar levels, which are usually reported to have worse prognoses [9,12,15].

Very few studies have reported on the treatment and characteristics of patients with iSFTs at diagnosis and at relapse [9,15,16]. To our knowledge, we carried out the first description of the clinical management of relapses in France and the largest one in Europe and in the USA. Recurrences occur mostly at the initial local site and are belated and repetitive [23]. Like Rutkowski et al., we observed that the treatment of the first relapse consisted of local techniques (radiotherapy and surgery) for most of our patients [9]. In concordance with the literature, we observed that the first relapse represented a turning point in iSFT evolution, with reduced recurrence latency over the course of the disease [9,12,15,25].

The specificity of iSFT is its potential for extracranial metastatic spreading in approximately 11% to 60% of cases [1,2,15]. The effectiveness of systemic treatments appears limited [28,29,30]. Nevertheless, antiangiogenic targeted therapy has shown promising OS and PFS results [8,28,31]. Interestingly, we noticed a long-term recurrence-free survival after the fourth local recurrence in one patient treated using an INF-alpha inhibitor. This observation strengthens the rationale of an influence of angiogenesis inhibition in treating unresectable iSFTs [11,32,33].

Although our cohort consisted of the largest description of iSFT cases in Europe and the USA, several limitations remained, especially those inherent in their retrospective nature. For example, some aspects could not be monitored in our study, especially those related to neuropsychiatric symptoms. Only one iSFT was declared to be revealed with psychiatric relapse, and some others with mental confusion regrouped in high-function disorders, but no psychiatric-specific tests were routinely performed during their surveillance. Nowadays, iSFT diagnosis requires the STAT6 immunohistochemistry status [4,5]. Another limitation of our study was the STAT6 status. The period of inclusion of our study was prior to the current use of this tool, explaining that we had less than 40% of patients with a known STAT6 status in our cohort, but all were positive. Nevertheless, our patients were included if there was a confirmed pathological review of iSFT specimens; otherwise, the patient was excluded (*n* = 6). At relapse, all specimens tested for STAT6 in immunochemistry were positive. Moreover, no difference for patients’ characteristics was seen in our cohort depending on a known STAT6 status. We conducted a long-term study that enabled extended surveillance to detect potential late recurrence. However, the time and period of surveillance were not consensual in France [1,9]. This might explain why some of our patients were lost-to-follow-up before 10 years of surveillance. An underestimation of recurrence and metastasis might have occurred.

## 5. Conclusions

Intracranial SFTs are rare sarcomas, which must be followed closely and for a long time after multimodal treatment because of their slow progression. From this study, there emerged the fact that iSFT management is heterogeneous within and between centres, particularly for the treatment of repeated recurrences.

Potential clinical, biological, and radiological factors predicting recurrence require further in-depth studies, especially at an international level. Moreover, prospective studies are necessary to evaluate new modalities of treatment after GTR or suboptimal resection, such as a new regimen of chemotherapy or refined modern techniques, such as proton therapy.

## 6. Suggested Recommendations Resulting from Our Clinical Experience

Intracranial SFTs should first be treated with surgical techniques. Adjuvant radiotherapy seemed to reduce recurrence in patients with a high histological grade and incomplete resection. Its indication could be discussed in multidisciplinary meetings. Local recurrence and oligometastatic diseases should be treated with local techniques (surgery, SRS) if feasible. Finally, systemic treatments lack efficiency, and clinical studies should be proposed.

## Figures and Tables

**Figure 1 cancers-15-00704-f001:**
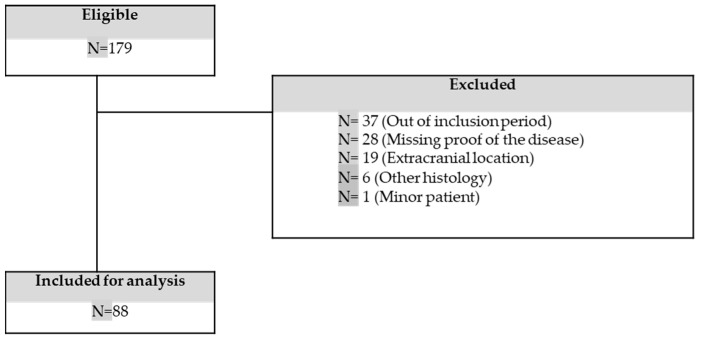
Flow chart of the study.

**Figure 2 cancers-15-00704-f002:**
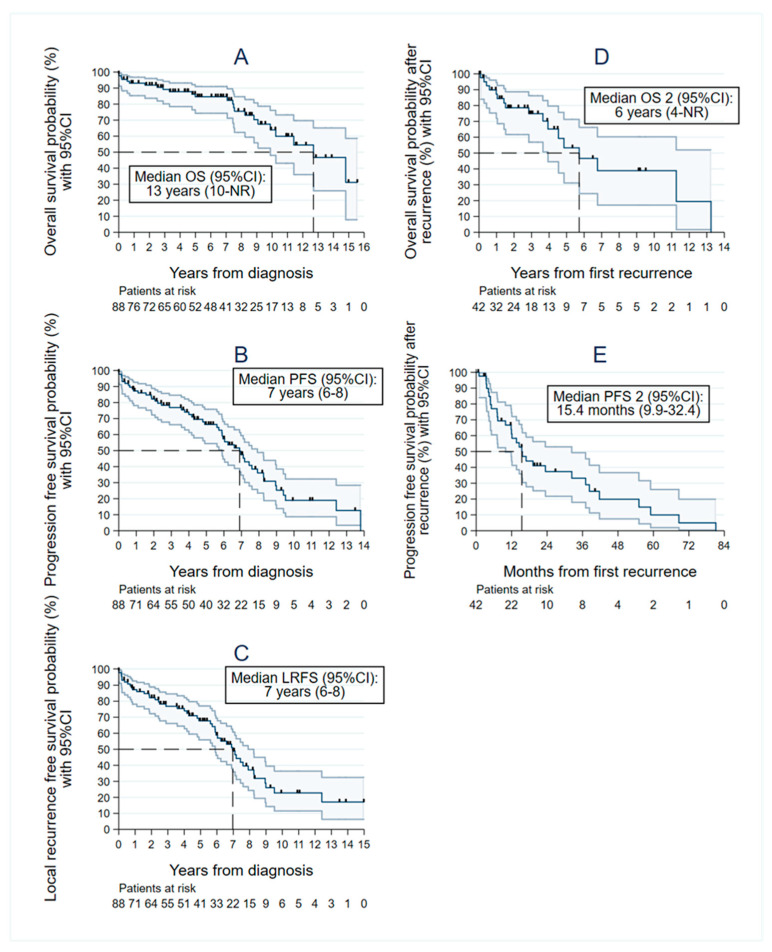
Survival analysis in newly diagnosed and recurrent iSFT patients. (**A**) Survival curve according to the Kaplan–Meier method showing median OS, (**B**) PFS, and (**C**) LRFS in newly diagnosed iSFTs and (**D**) OS2 and (**E**) PFS2 after the first recurrence, with their respective 95% confidence intervals (95% CI). NR: nonreached.

**Figure 3 cancers-15-00704-f003:**
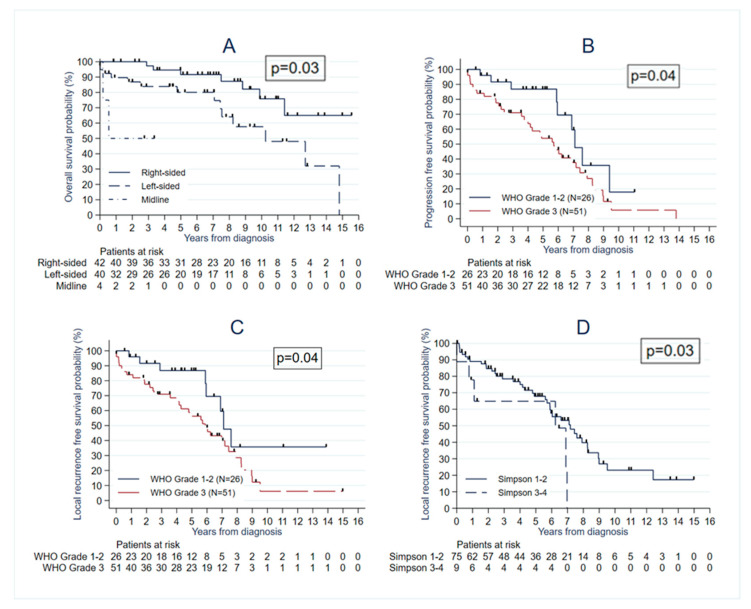
The survival curve obtained with the Kaplan–Meier method showing the difference in OS according to tumour location (**A**), PFS according to the WHO grade (**B**), LRFS according to the WHO grade (**C**), and LRFS according to the extent of resection with newly diagnosed iSFT (**D**) with their respective p value, significant if <0.05.

**Table 1 cancers-15-00704-t001:** Clinical, pathological, morphological, and treatment characteristics for the 88 patients presenting a new diagnosis of iSFT.

Characteristics (*N* = 88)	*n* (%)
**Sex**	
Female	50 (57%)
Male	38 (43%)
**Median age (range)**	54.5 (19–88)
Mean age (SD)	51.5 (16.4)
**STAT6 expression (immunohistochemistry)**	
Yes	34 (39%)
No	0 (0%)
Not available	54 (61%)
**The WHO histoprognostic grade**	
Grade 1	1 (1%)
Grade 2	25 (28%)
Grade 3	51 (58%)
Not available	11 (13%)
**Mitotic count**	
<5	36 (41%)
≥5	45 (51%)
Not available	7 (8%)
**Necrosis**	
Yes	33 (38%)
No	55 (63%)
**Number of lesions**	
Unique	84 (95%)
Multiple	4 (5%)
**Maximal diameter**	
< 3cm	5 (6%)
3–5 cm	24 (27%)
≥5 cm	32 (36%)
Not available	27 (30%)
**Topography of the tumour**	
Right-sided	42 (48%)
Left-sided	40 (45%)
Midline *	4 (4%)
Bilateral	1 (1%)
Not available	1 (1%)
**Location**	
Supratentorial	71 (81%)
Infratentorial	17 (17%)
Supra- and infratentorial	2 (2%)
**Preoperative embolization**	
Yes	8 (9%)
No	80 (91%)
**Resection grade according to Simpson**	
Simpson 1–2 (gross tumour resection)	75 (85%)
Simpson 3–4 (subtotal tumour resection)	9 (10%)
Simpson 5 (biopsy)	1 (1%)
Not available	3 (3%)
**Postoperative radiotherapy (*N* = 56)**	
Yes	32 (57%)
No	24 (43%)
**Technique of radiotherapy**	
Conformational radiotherapy Radiosurgery Not available	19 (50%) 3 (9%) 10 (31%)
**Local control**	
Yes	76 (86%)
No	11 (13%)
Not available	1 (1%)
**Type of treatment (*N* = 65)**	
Surgery alone	33 (51%)
Radiotherapy alone	1 (1%)
Surgery plus postoperative radiotherapy	31 (48%)

* Falx and pineal region tumours.

**Table 2 cancers-15-00704-t002:** Morphological and therapeutic management data in patients with at least one episode of localized recurrence of iSFT.

Characteristics (*N* = 37)	*n* (%)
**Unifocal recurrence**	
Yes	32 (87%)
No	5 (14%)
**Maximal diameter of recurrence**	
<3 cm	19 (51%)
3–5 cm	5 (14%)
≥5 cm	2 (5%)
Not available	11 (30%)
**Recurrence topography**	
Right-sided	18 (49%)
Left-sided	16 (43%)
Midline	3 (8%)
Bilateral	0 (0%)
**Recurrence location**	
Supratentorial	30 (81%)
Infratentorial	7 (19%)
Supra- and infratentorial	0 (0%)
**Preoperative embolization**	
Yes	0 (0%)
No	37 (100%)
**Preoperative radiotherapy**	
Yes	1 (3%)
No	36 (97%)
**Surgery**	
Yes	17 (46%)
No	20 (54%)
**Resection grade according to Simpson (*N* = 17)**	
Simpson 1–2 (gross tumour resection)	15 (88%)
Simpson 3–4 (subtotal tumour resection)	1 (6%)
Not available	1 (6%)
**Postoperative radiotherapy (*N* = 17)**	
Yes	10 (59%)
No	7 (41%)
**Type of radiotherapy**	
Conformational radiotherapy Radiosurgery Not available	4 (40%) 4 (40%) 2 (20%)
**Exclusive radiotherapy**	
Yes	14 (38%)
No	23 (62%)
**Technique of radiotherapy**	
Conformational radiotherapy Radiosurgery Not available	2 (14%) 8 (57%) 4 (29%)
**Chemotherapy associated with radiotherapy**	
Yes	1 (3%)
No	36 (97%)
**Postoperative chemotherapy**	
Yes	1 (3%)
No	36 (97%)
**Local control after treatment**	
Yes	26 (70%)
No	10 (27%)
Not available	1 (3%)
**Type of treatment that led to control (*N* = 26)**	
Surgery alone	4 (15%)
Radiotherapy alone	11 (42%)
Surgery and postoperative radiotherapy	11 (42%)
Chemotherapy alone	0 (0%)
Radiotherapy and postoperative chemotherapy	0 (0%)
No treatment	0 (0%)

**Table 3 cancers-15-00704-t003:** Univariate and multivariate analyses according to Cox models in patients presenting an initial diagnosis of iSFT.

Characteristics	Overall Survival				Progression-Free Survival				Local Recurrence-Free Survival			
	Univariate HR (95% CI)	*p*	Multivariate HR (95% CI)	*P*	Univariate HR (95% CI)	*p*	Multivariate HR (95% CI)	*P*	Univariate HR (95% CI)	*p*	Multivariate HR (95% CI)	*p*
Sex		0.78				0.36				0.60		
Female	1				1				1			
Male	0.89(0.39–2.02)				0.76 (0.43–1.36)				0.85(0.48–1.53)			
**Age ***	**1.05(1.02–1.09)**	**0.003**	**1.04(1.01–1.08)**	**0.01**	**1.03 (1.01–1.05)**	**0.003**	**1.03(1.01–1.05)**	**0.008**	**1.03(1.01–1.05)**	**0.002**	**1.04(1.01–1.06)**	**0.002**
Signs of raised intracranial pressure ^$^ (*n* = 85)	1.30(0.53–3.17)	0.57			0.95(0.48–1.89)	0.89			0.99(0.50–1.97)	0.98		
Motor deficit ^$^ (*n* = 85)	1.56 (0.63–3.84)	0.34			1.34 (0.71–2.51)	0.37			1.28 (0.67–2.44)	0.46		
Sensory deficit ^$^ (*n* = 85)	1.11 (0.26–4.80)	0.89			0.90 (0.32–2.51)	0.84			0.95 (0.34–2.66)	0.92		
Epileptic seizures *^$^ (*n* = 85)	0.22 (0.03–1.66)	0.14			0.37 (0.13–1.03)	0.056			0.40 (0.14–1.11)	0.08		
High-function disorder *^$^ (*n* = 85)	1.65 (0.72–3.78)	0.23			1.52 (0.86–2.70)	0.15			1.52 (0.85–2.73)	0.16		
Visual disorder ^$^ (*n* = 85)	0.60 (0.20–1.77)	0.35			0.95 (0.50–1.81)	0.88			1.02 (0.53–1.95)	0.95		
Cerebellar syndrome *^$^ (*n* = 86)	1.83 (0.67–5.00)	0.24			1.78 (0.83–3.86)	0.14			1.86 (0.86–4.02)	0.12		
Headache ^$^ (*n* = 85)	1.00 (0.43–2.33)	1.00			0.97 (0.54–1.73)	0.91			0.83 (0.46–1.50)	0.54		
Mitotic count ≥5 *^$^ (*n* = 81)	1.47 (0.62–3.48)	0.38			**2.03 (1.07–3.85)**	**0.03**			**2.36 (1.21–4.63)**	**0.01**		
Necrosis ^$^	1.25 (0.56–2.81)	0.59			1.15 (0.65–2.06)	0.63			1.15 (0.64–2.07)	0.65		
**WHO grade** (*n* = 77)		0.29				0.047		0.04		0.03		0.03
**Grade 1–2**	1				**1**				1			
**Grade 3**	1.76 (0.58–5.28)				**2.05 (0.98–4.28)**		**2.14 (1.03–4.78)**		2.21 (1.02–4.79)		2.36 (1.08–5.16)	
Max. diameter (*n* = 61)		0.83				0.79				0.74		
< 3 cm	1				1				1			
3–5 cm	1.73(0.18–228.92)				1.96(0.25–15.16)				1.94(0.25–15.00)			
≥5 cm	2.20(0.27–285.32)				1.70(0.22–12.93)				1.55(0.20–11.85)			
**Topography ***(*n* = 87)		**0.004**		**0.03**		**0.003**				**0.002**		
Right-sided	**1**		**1**		**1**				**1**			
**Left-sided ***	**3.02(1.21–7.52)**		**2.69(1.07–6.81)**		**1.84 (1.01–3.37)**				**2.03 (1.10–3.76)**			
**Midline ***	**13.86(2.54–75.71)**		**8.29(1.42–48.50)**		**8.57(2.32–31.65)**				**9.08(2.44–33.73)**			
**Location**		0.94				0.70				0.62		
Supratentorial	1				1				1			
Infratentorial	0.96 (0.32–2.84)				1.15 (0.55–2.40)				1.20 (0.58–2.51)			
Preoperative embolization *^$^	0.25 (0.002–1.82)	0.35			**0.08 (0.001–0.55)**	**0.08**			**0.09 (0.001–0.60)**	**0.09**		
**Resection grade *** (*n* = 84)		**0.04**				**0.12**				**0.10**		**0.03**
**Simpson 1–2**	1				**1**		1		1		1	
**Simpson 3–4**	**3.20 (1.04–9.83)**				**2.02 (0.84–4.89)**				**2.11 (0.87–5.12)**		**3.00 (1.09–8.24)**	
PORT ^$^ (*n* = 56)	0.88 (0.34–2.29)	0.80			0.50 (0.26–0.97)	0.04			0.56 (0.29–1.10)	0.10		

* Univariate significant parameters (*p* < 0.20) were then tested in a multivariate model; they were significant if *p* < 0.05, indicated by bolded text. HR: hazard ratio; PORT: postoperative radiotherapy; ^$^ For binary variables, the reference category was “No”.

## Data Availability

The data presented in this study are available on request from the corresponding author.

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
