# Peer review of "Intracranial Solitary Fibrous Tumour Management: A French Multicentre Retrospective Study"

_cancers, 2023, doi:10.3390/cancers15030704_

Round 1

Reviewer 1 Report

Submitted manuscript concerns a multicenter study regarding hemangiopericytoma course, relapse mechanisms and outcome of interventions in France. In general, the manuscript is well presented and provides a lot of relevant information enabling further development of effective therapeutic interventions in the analyzed nervous system tumor. The study includes a relatively high number of preselected patients.

However, I believe that several aspects could be extended and improved.

- please compare the specificity of French conditions and procedures with other EU and non-EU countries. The point is to provide information in a few sentences that allow the readers to discover how universal or unique the different procedures and rules of conduct are.

- taking into account that various diagnostic and therapeutic procedures are related to cooperation with the patient (e.g. in the form of informed consent), was the presence of clinical dementia (not just an overall cognitive status of the patients) monitored, and were any other neuropsychiatric symptoms (e.g. depressive) taken into account in the qualification?

- Were possible occurrences of other oncological diseases and tumors located outside the nervous system monitored? If so, what diseases and where exactly were they? (I'm not talking about metastases from the primary cancer)

- in terms of the analyzed predictors, whether there were correlations between them should be analyzed.

- what unique data and conclusions, in this case, result from the multi-center study, and what new recommendations can be suggested to other centers? It would be worth separating a separate subchapter regarding potential practical conclusions and recommendations resulting from the clinical experience of the authors of the manuscript.

Author Response

Submitted manuscript concerns a multicenter study regarding hemangiopericytoma course, relapse mechanisms and outcome of interventions in France. In general, the manuscript is well presented and provides a lot of relevant information enabling further development of effective therapeutic interventions in the analyzed nervous system tumor. The study includes a relatively high number of preselected patients.

However, I believe that several aspects could be extended and improved.

- please compare the specificity of French conditions and procedures with other EU and non-EU countries. The point is to provide information in a few sentences that allow the readers to discover how universal or unique the different procedures and rules of conduct are.

Thank you for your suggestion. We clarified our introduction: “Surgery is the universal mainstay of SFT treatment[...] However, PORT’s impact on the overall survival (OS) of iSFT is still discussed leading to various managements at an international and a national levels”

- taking into account that various diagnostic and therapeutic procedures are related to cooperation with the patient (e.g. in the form of informed consent), was the presence of clinical dementia (not just an overall cognitive status of the patients) monitored, and were any other neuropsychiatric symptoms (e.g. depressive) taken into account in the qualification?

Thank you for your interesting comment. The patients included in the study were those who gave their consent to be operated. However, we have not recorded a refusal of adjuvant radiotherapy. Concerning the neuropsychological symptoms, we had poor information, and we added a sentence concerning that limit in the discussion: “Only one iSFT was revealed by psychiatric symptoms and some others were discovered with mental confusion who were classified in the category “high function disorders” but no psychiatric specific tests were done in routine during their surveillance.”

- Were possible occurrences of other oncological diseases and tumors located outside the nervous system monitored? If so, what diseases and where exactly were they? (I'm not talking about metastases from the primary cancer): We collected the medical background of each patients including cancers however no other solid or haematological cancer occurred to our knowledge. We added this information in the manuscript: “To our knowledge, no other solid or hematological cancer occurred during the time of surveillance. “

- in terms of the analyzed predictors, whether there were correlations between them should be analyzed. Thank you for your comments. Correlations between variables were searched.  This sentence has been added in manuscript in the methodology part. To detail, only 2 variables for OS and PFS and three variables for LRFS were included in the final model of which just one was quantitative (age). No correlation test of spearman were done. A contingency table was done to analyse if the distribution of the WHO grade was dependent on the resection grade. The exact test of fischer was not significant. (p=0.314)

QT

QL

Correlation test

OS

Age

Tumour topography

-

PFS

Age

WHO grade

LRFS

Age

WHO grade

WHO grade * Resection grade according to Simpson

Fisher's exact test(p=0.314)

- what unique data and conclusions, in this case, result from the multi-center study, and what new recommendations can be suggested to other centers? It would be worth separating a separate subchapter regarding potential practical conclusions and recommendations resulting from the clinical experience of the authors of the manuscript.

 Thank you for this suggestion, we added our suggestions in a paragraph after the conclusion: “Intracranial SFT should be first treated by surgical techniques. Adjuvant radiotherapy seems to reduce recurrence in patients with high histological grade and uncomplete resection. Its indication might be discussed in multidisciplinary meeting. Local recurrence and oligometastatic diseases should be treated by local techniques (surgery, SRS) if feasible. Finally, systemic treatment lack efficiency and clinical study should be proposed. »

Reviewer 2 Report

Lottin et al report a multicenter retrospective review of 88 French patients with intracranial solitary fibrous tumors. GTR was found to be associated with progression free survival, but radiotherapy was not. The authors had an impressive median follow up time of 7 years and median OS of 13 years. Recurrence occurred in half of patients. Most were supratentorial. This is a sizable study and worthwhile to publish.

Author Response

Thank you very much. We are grateful for your review.

Reviewer 3 Report

The authors analyzed survival outcomes of intracranial solitary fibrous tumors in French brain tumor database. Eighty-eight patients were followed up for median 7 years. The authors reported that gross total removal was a prognostic factor for PFS, but not WHO grade nor radiotherapy. Surprisingly, the side of the tumors was a strong predictor for OS.

The most serious problem was incomplete pathological evaluation. WHO grade and STAT6 expression were not evaluated in 30% and 39%, respectively. Furthermore, their slides did not undergo central review.

It may be possible to find WHO 2021 classification because mitotic count and necrosis were recorded. At that time, necrosis by preoperative embolization should be distinguished.

Although the authors described that radiotherapy was not effective, they did not analyze in subgroups. Radiotherapy might be applied to patients with a residual tumor or histologically aggressive tumor.

Type of radiotherapy should be described (SRS, conventional radiotherapy, or brachytherapy).

What was the definition of “midline tumor”? Did it include parasagittal or falx tumors?

In page 11 line 289-290,

----in multivariate analysis (HR=0.08 [95%CI: 0.001-0.55; p=0.08] and HR=0.09 [95%CI: 0.001-0.60; p=0.09]) 

Are these descriptions correct? While upper 95%CIs were much smaller than 1, p value exceeded 0.05. These values are cited as univariate analysis in Table 3.

Resection grade was a prognostic factor of OS in univariate analysis (p=0.04), but the result in multivariate analysis was not recorded in Table 3. I wonder if this was caused by a stepwise method in multivariate analysis. Some of statisticians criticize a stepwise method because it sometimes skews results. Have you consulted a statistician?

Author Response

The authors analyzed survival outcomes of intracranial solitary fibrous tumors in French brain tumor database. Eighty-eight patients were followed up for median 7 years. The authors reported that gross total removal was a prognostic factor for PFS, but not WHO grade nor radiotherapy. Surprisingly, the side of the tumors was a strong predictor for OS.

The most serious problem was incomplete pathological evaluation. WHO grade and STAT6 expression were not evaluated in 30% and 39%, respectively. Furthermore, their slides did not undergo central review.

It may be possible to find WHO 2021 classification because mitotic count and necrosis were recorded.

We thank you for your relevant comments concerning pathology. We actualized our analyses according to the 2021 CNS WHO grade classification and better specify the limit of STAT6 expression in discussion: “Another limit of our study is STAT6 status. The period of inclusion of our study was prior to the current use of this tool, explaining that we had less than 40% of patients with a known STAT6 status in our cohort, but all positive. Nevertheless, our patients were included if there was a confirmed pathological review of iSFT specimens, otherwise the patient was excluded (n=6).”

Concerning the CNS grade, our update led to confirm the prognosis factor of CNS grade for PFS and LRFS. All the results were updated.: “Higher histological grade was significantly associated with lower PFS and LRFS (HR=2.14 [95%CI: 1.03-4.78; p=0.04] and HR=2.36 [95%CI: 1.08-5.16; p=0.03], respectively) (Table 3).

At that time, necrosis by preoperative embolization should be distinguished. Confirmed and sentence deleted.

Although the authors described that radiotherapy was not effective, they did not analyze in subgroups. Radiotherapy might be applied to patients with a residual tumor or histologically aggressive tumor.

Thank you for this important comment. We updated our analysis with a subgroup concerning those patients (n=56).

The univariate analysis were then updated. Although p<0.20 for PFS and LFS, this covariate was not included in the principal final multivariate model, because it concerns a subgroup of patients.

PORT (N=56)

0.80

OS

No

7/24

1 (ref)

Yes

11/32

0.88 (0.34-2.29)

PORT (N=56)

0.04

PFS

No

17/24

1 (ref)

Yes

21/32

0.50 (0.26-0.97)

PORT (N=56)

0.10

LFS

No

16/24

1 (ref)

Yes

20/32

0.56 (0.29-1.10)

All results had been updated in the main manuscript : “Adjuvant radiotherapy was significantly associated with greater PFS and LRFS in univariate analysis (HR=0.50 [95%CI: 0.26-0.97; p=0.04] and HR=0.56 [95%CI: 0.29-1.10]; p=0.10]), respectively, but not with OS (HR= 0.88 [95%CI: 0.34-2.29); p=0.80]). Sensitivity multivariate analyses were performed with PORT as a covariate. We observed a significantly lower PFS with HR=0.47 (95%CI: 0.24-0.91; p=0.03), but the association was no more significant regarding LRFS.

Type of radiotherapy should be described (SRS, conventional radiotherapy, or brachytherapy).

The description of the type of radiotherapy was added to table 1 and 2

What was the definition of “midline tumor”? Did it include parasagittal or falx tumors? Midline tumors included falx and pineal region tumors

In page 11 line 289-290,

----in multivariate analysis (HR=0.08 [95%CI: 0.001-0.55; p=0.08] and HR=0.09 [95%CI: 0.001-0.60; p=0.09]) 

Are these descriptions correct? While upper 95%CIs were much smaller than 1, p value exceeded 0.05. These values are cited as univariate analysis in Table 3.

The Hazard-Ratio for embolization was obtained by a Cox model with Firth correction given the very low number of events (0) in the category Yes (Software: SAS Enterprise Guide 8.3). After checking, those p-values are correct and the inconsistency between CI95% and p-value came from this correction which is justified by a modality without events.

PFS

Nb evts/obs

HR univariate (IC95%)

p

Pre operative Embolisation

0.08

No

49/80

1 (ref)

Yes

0/8

0.08 (0.001-0.55)

LRFS

Nb evts/obs

HR univariate (IC95%)

p

Pre operative Embolisation

0.09

No

46/80

1 (ref)

Yes

0/8

0.09(0.001-0.60)

Resection grade was a prognostic factor of OS in univariate analysis (p=0.04), but the result in multivariate analysis was not recorded in Table 3. I wonder if this was caused by a stepwise method in multivariate analysis. Some of statisticians criticize a stepwise method because it sometimes skews results. Have you consulted a statistician?

The grade of Simpson was not significant in the multivariate model construction with a step-by-step top-down procedure. The robustness of this model was confirmed and the lack of significativity was confirmed. Each variable significant in univariate analysis was then re-integrated one by one to check the validity of the model. We confirmed that resection grade was not significant: "please see attachment"

Round 2

Reviewer 3 Report

The authors have responded adequately. I have no further comments.